# Fertility Control for Wildlife: A European Perspective

**DOI:** 10.3390/ani13030428

**Published:** 2023-01-27

**Authors:** Giovanna Massei

**Affiliations:** Botstiber Institute for Wildlife Fertility Control Europe, Department of Environment and Geography, University of York, 290 Wentworth Way, Heslington, York YO10 5NG, UK; giovanna.massei@york.ac.uk

**Keywords:** coexistence, contraceptives, human–wildlife conflicts, wildlife management, wildlife impacts

## Abstract

**Simple Summary:**

Current trends of human population growth and landscape development in Europe suggest that wildlife impacts will increase. Traditional methods to mitigate these impacts, such as culling, can be ineffective, environmentally harmful and often publicly opposed. Contraceptives might be an alternative to culling. This review focuses on contraceptives for mammals and birds, currently registered or widely tested, which might be considered for European wildlife. The review describes the effects of contraceptives on reproduction and welfare, the methods and challenges of contraceptive administration to large numbers of animals, the potential cost and feasibility of using fertility control and the knowledge gaps in this area. Contexts and species for using contraceptives to reduce the impacts of European wildlife include small, isolated wildlife populations, charismatic species and situations in which lethal control is either illegal or publicly unacceptable, such as urban environments and national parks. The review provides guidance to assist decisions about the potential use of wildlife fertility control and lists eight main reasons for Europe to invest in this area, and particularly in developing oral contraceptives which will allow large-scale applications of wildlife fertility control. This would be crucial for Europe, where humans and wildlife are increasingly sharing space and resources.

**Abstract:**

Trends of human population growth and landscape development in Europe show that wildlife impacts are escalating. Lethal methods, traditionally employed to mitigate these impacts, are often ineffective, environmentally hazardous and face increasing public opposition. Fertility control is advocated as a humane tool to mitigate these impacts. This review describes mammalian and avian wildlife contraceptives’ effect on reproduction of individuals and populations, delivery methods, potential costs and feasibility of using fertility control in European contexts. These contexts include small, isolated wildlife populations and situations in which lethal control is either illegal or socially unacceptable, such as urban settings, national parks and areas where rewilding occurs. The review highlights knowledge gaps, such as impact of fertility control on recruitment, social and spatial behaviour and on target and non-target species, provides a decision framework to assist decisions about the potential use of wildlife fertility control, and suggests eight reasons for Europe to invest in this area. Although developing and registering contraceptives in Europe will have substantial costs, these are relatively small when compared to wildlife’s economic and environmental impact. Developing safe and effective contraceptives will be essential if European countries want to meet public demand for methods to promote human–wildlife coexistence.

## 1. Introduction

Current trends of human population growth and landscape development in Europe suggest that human–wildlife interactions will multiply, as more people and wildlife share the same spaces and resources. When wildlife threatens human and animal health and safety, the food supply or the environment, these interactions become conflicts [1]. The economic and environmental impacts of wildlife include damage to crops and forestry, disease transmission, vehicle collisions, impact on biodiversity, livestock predation and attacks on people. Many of these impacts are due to local populations of wildlife exceeding the so-called “social carrying capacity”: in these instances, wildlife is referred to as “overabundant” [2,3,4,5].

In Europe, the escalation of wildlife impacts is due to many factors such as changes in farm management, abandonment of farmlands and increase in forested areas, suburban sprawl, decreasing number of hunters, return of large carnivores, introductions of invasive non-native species and increased density of people [6,7,8,9]. Examples of overabundant wildlife in Europe include wild boar (*Sus scrofa*) [7], several deer species [4,8] and rodents such as Norway rats (*Rattus norvegicus*) and common voles (*Microtus arvalis*) [10,11]. Many non-native mammals and birds, defined as invasive alien species (IAS) of European concern are also regarded as overabundant. These comprise several species of squirrels and mongooses, coypu (*Myocastor coypus*), raccoons (*Procyon lotor)*, Canada geese (*Branta canadensis*) and parakeets [12,13,14,15].

Examples of environmental and economic impacts of overabundant wildlife in Europe range from vole outbreaks to transmission of diseases to other wildlife, livestock and people, road traffic accidents with ungulates and airstrikes. For instance, in a vole outbreak year, up to 80% of alfalfa primary production in Poland was consumed by common voles, and in Germany financial losses due to crop damage amounted to hundreds of millions of euros [16]. In England alone, the cost of controlling bovine tuberculosis for 10 years in cattle and European badgers (*Meles meles*) was estimated as 500 million British pounds [17]. In Spain, 8.9% of the 74,600 road traffic accidents reported between 2006 to 2012 were due to wildlife vehicle collisions, with wild boar and roe deer (*Capreolus capreolus*), involved in 79% of instances and with a cost of EUR 105 million per year [18]. 

Traditional methods used to mitigate wildlife impacts include culling and toxicants, with many millions of wildlife killed every year in Europe [6,7]. For instance, in 67.9% of the 209 European national parks in 29 countries, wildlife is regulated by culling, hunting or both [19], and the Europe rodenticide market, worth USD 250 million in 2021, is projected to grow to USD 307.1 million in 2026 (https://www.marketdataforecast.com/market-reports/europe-rodenticides-market accessed 15 December 2022). In parallel, public opposition to culling is increasing, mainly driven by animal welfare concerns, human safety in urban settings, environmental impact of some lethal methods and a lack of efficacy of culling in addressing impacts or reduction in wildlife population sizes [20,21,22]. In some countries the environmental impact of second-generation rodenticides on predators and scavengers has already prompted public authorities to consider banning or restricting their use [23,24,25]. Consequently, there has been interest in non-lethal methods such as fertility control, increasingly advocated as a safe and humane means of managing overabundant wildlife [26,27,28,29]. This interest has grown particularly in Western countries, where most citizens now live in urban areas, have very little direct contact with wildlife, but often express strong views against lethal control, compared to people living in rural settings that have a more utilitarian approach to wildlife [30,31]. This is especially noticeable in Europe, where the current density of the human population (34/km^2^), is almost double the density in the US (20/km^2^) and where it is estimated that, by 2050, 80% of people will live in urban areas (https://population.un.org/wpp/Download/Standard/MostUsed/ accessed 15 December 2022).

In the European Union (EU) wildlife management is regulated by international standards such as the Agreement on International Humane Trapping Standards, by EU law (Regulation No 1143/2014 on the prevention and management of the introduction and spread of invasive alien species) and by national legislation. The latter covers specific issues such as culling, as a means of prevention of infectious diseases, or the control of animals with negative economic or environmental impacts. Whilst a few documents mention the use of wildlife fertility control in European countries [32,33], mainly in the context of management of invasive non-native species, little attention has been paid so far to the applicability of this method in different contexts and for different wildlife species. 

The aims of the report are: 1. to provide a brief update on contraceptives, delivery methods, contexts and wildlife species for which fertility control has been used, with particular focus on European species; 2. to offer a logical structure for guiding decisions on practical applications of fertility control to mitigate wildlife impacts; and 3. to highlight knowledge gaps and areas of investment in wildlife fertility control for European Member States so that the potential of this tool can be fully realised and put into practical applications.

## 2. Wildlife Contraceptives

This overview includes only contraceptives that have been tested in free-living birds and mammals but excludes fertility control drugs that are commonly employed in controlled conditions, such as in zoos or that are used mainly on companion animals such as cats and dogs. The latter include hormonal implants which require a minor surgical procedure by a veterinarian. Zoos and owners of companion animals have the advantage of being able to select contraceptive type, doses and frequency of administration, and to monitor the effect of these drugs on reproduction at any time, which is clearly not the case for free-living wildlife. The overview also does not include surgical sterilization, which has been used in isolated cases such as during the eradication of the Eastern grey squirrel from an urban park in Italy [34], nor technologies such as gene-drive, gene-transfer or oral contraceptive vaccines, that are still at an early stage of development.

### 2.1. Immunocontraceptive Vaccines

Many studies of fertility control for wildlife have focussed on immunocontraceptives which target hormones or proteins essential for reproduction and are available as injectable vaccines. The most frequently used immunocontraceptives, developed for mammals, are those based on the gonadotropin-releasing hormone (GnRH) and on zona pellucida (ZP) proteins [26,27,35]. Immunocontraceptives can prevent ovulation, sperm production or fertilisation and are typically coupled with adjuvants, which are compounds used to amplify the immune response to a vaccine. The effectiveness, longevity and side effects of immunocontraceptive vaccines depend on species, gender, age, as well as on the active compound, its formulation, delivery route and type of adjuvant [22,26,29].

The zona pellucida (ZP) is a layer of proteins which surrounds an ovulated egg and allows sperm recognition and binding. Porcine zona pellucida (PZP)-based immunocontraceptive vaccines stimulate the production of antibodies that bind to sperm receptors on the egg’s surface, thus preventing sperm attachment and fertilization. PZP-based injectable vaccines have been effective in females of many ungulate species, seals and bears, but not in rodents and wild pigs [27,29,36]. 

PZP-based immunocontraceptives are available in several formulations and have been used with various adjuvants. The most commonly used PZP-based contraceptives are: i) native PZP such as ZonaStat-H and ZonaStat-D, registered in the USA for use in wild equids and cervids, respectively; ii) PZP-22, based on a single dose of native PZP plus controlled-release microspheres that release PZP at 1, 3 and 12 months following the first injection, mimicking a series of booster doses; iii) SpayVac^®^, based on an adjuvant and PZP encapsulated within vesicles called liposomes, which gradually release PZP over an extended period of time.

The multi-year effectiveness of PZP-based contraceptives depends on these vaccines’ formulation. For instance, Killian et al. [37] showed that single doses of SpayVac^®^ significantly reduced fertility in a small sample of captive horses (*Equus caballus*) for up to 4 years. Gray et al. [38] found that a formulation of PZP with an adjuvant called Adjuvac reduced fertility in free-roaming mares for 3 years, although efficacy was lower than in captive studies. A single injection of SpayVac rendered infertile horses, feral donkeys (*Equus asinus*) and white-tailed deer (*Odocoileus virginianus*) for multiple years [39,40,41,42]. Turner et al. [43] showed that reproductive success in mares treated with a single dose of PZP-22 induced 2–3 years of substantially reduced fertility, followed by a return to a control level of fertility in 4 years [43]. However, attempts to replicate this multi-year effect in mares at three other study areas in the USA had a lower success, possibly due to differences in the composition and manufacturing of the vaccine’s components [44,45]. 

In Europe a PZP-based vaccine has been used in a free ranging feral horse population in the Romanian Danube Delta in a study to test the feasibility of administering this contraceptive manually to individual mares, with subsequent remote booster delivered by dart gun. Mares in the treatment group were vaccinated and 50% were administered a booster dose one year later. The following year, the pregnancy rate of the PZP treated mares was 14.6% compared to 81.5% in the control group, although 44% of the mares did not receive a booster injection [46]. The study concluded that a PZP contraception program through individual immobilization followed by remote booster administration is feasible, although the approach is time and resource consuming

Negative effects of ZP vaccines, observed in some species but not in others, include multiple infertile oestrous cycles leading to extended breeding season, increased movements, potential late births and disruption of social hierarchy [27,47,48,49,50]. Other studies on deer and feral horses have found that ZP-based vaccines did not affect time budget, social behaviour and body condition [35,51,52] and that these contraceptives increased animals’ lifespan [53]. PZP is safe to administer to pregnant or lactating females [54,55]. Injection-site reactions such as abscesses and granulomas (thickened tissue filled with fluid)) may occur in animals such as horses and white-tailed deer treated with ZP vaccines, but there is no evidence that these reactions are painful or affect mobility [38,42,48]. 

Gonadotrophin Releasing Hormone (GnRH) immunocontraceptive vaccines stimulate an animal’s immune system to create antibodies against the GnRH, which results in decreased concentrations of sex hormones and inhibits reproduction in both sexes [56]. The GnRH-based vaccine most studied in wildlife is GonaCon, registered in the USA as a contraceptive for white-tailed deer, feral horses, feral donkeys and prairie dogs (*Cynomys ludovicianus*). Formulated as an injectable synthetic GnRH coupled to a mollusk protein and to the adjuvant AdjuVac [56], GonaCon induced infertility for several years in deer, wild boar, horses, feral cattle (*Bos taurus* and *Bos indicus*) and bison (*Bison bison*) after one or a few doses [37,38,56,57,58,59]. As GonaCon prevents ovulation, treated females and males do not exhibit reproductive behaviour. Although the longevity of effect depends on the species, in all instances booster vaccinations result in extended duration of infertility. For instance, a second booster dose administered to feral cattle 2–4 years after vaccination rendered all animals infertile for at least another year [59]. Similar results were found in free-ranging horses vaccinated with a single dose of GonaCon and re-immunised four years later [60]. After the first dose of GonaCon, the proportion of treated mares with foals was lower than that of control animals up to the third post-treatment year; a booster dose administered 4 years after the primer vaccination rendered between 84% and 100% of mares infertile for the following three years. 

In Europe, a captive study of wild boar in the UK showed that 11 out of 12 sows treated with GonaCon remained infertile for at least 4 to 6 years following a single injection, with no adverse effects on physiology, behaviour and welfare [61,62]. A subsequent pilot field study [63] showed that 4 of 5 GonaCon-treated sows, sampled between 9 and 30 weeks after a single injection, had antibody titres which indicated infertility. Another captive study in the UK on badgers found that vaccination of a single dose of GonaCon in early summer inhibited reproduction for at least one year [64]. Gonacon was also tested in Wales (UK) on a population of feral goats (*Capra hircus*) and the results suggested that vaccination of females significantly reduced their breeding success for two years [65].

In some species, such as white-tailed deer, vaccination with GonaCon caused a granuloma or a sterile abscess at the injection site [66] in most animals, although no evidence of limping or impaired mobility was observed. However, in fox squirrel (*Sciurus niger*) open abscesses, limping and stiff walking were recorded in some GonaCon-treated animals [67], but no differences were found in behaviour between vaccinated and untreated squirrels [68]. Injection site abscesses were also reported in Eastern grey squirrel (*Sciurus carolinensis*) [69]. Abnormal antler development after vaccination with GonaCon was observed in male white-tailed deer, leading to the conclusion that this contraceptive should not be used in antlered males [70]. Conversely, GonaCon had no adverse injection site reaction on wild boar and feral cattle and no adverse health effect on prairie dogs [61,62,71]. GonaCon does not affect existing pregnancies that go to term before an animal becomes infertile [57,63]. 

A key advantage of immunocontraceptive vaccines, compared to other contraceptives, is that these drugs are unlikely to affect predators or scavengers consuming treated animals, as the vaccines are destroyed in the gastro-intestinal tracts of the consumers [26]. The main disadvantage of immunocontraceptives is that these are only available as injectables, and thus of limited use for managing large numbers of wildlife, due to the cost involved in trapping and, in some cases, to the immobilization of animals to be treated.

### 2.2. Oral Contraceptives

At present the only fertility control drugs available for large-scale applications are two oral contraceptives developed for rodents and one oral contraceptive for birds. These are: ContraPest^®^, based on a combination of two active ingredients, 4-vinylcyclohexene diepoxide (VCD) and triptolide [72,73] and registered in the USA for black rats (*Rattus rattus*) and Norway rats; EP-1, which is a combination of two synthetic hormones, levonorgestrel and quinestrol, registered in Tanzania for multimammate mice (*Mastomys natalensis*) [74,75,76]; Ovistop, R-12 and Ovocontrol, all based on the active compound nicarbazin, and registered in Italy and Belgium (Ovistop and R-12) for urban pigeons (*Columbia livia*) and in the USA (Ovocontrol) for pigeons and other bird species [26,29,77]. 

ContraPest^®^ is a liquid contraceptive designed to reduce fertility in rats and delivered in a tray, placed inside a box, to minimise use by non-target species [78]. ContraPest suppresses fertility in males, by preventing sperm maturation and motility, and in females, by decreasing the number of eggs that are ovulated [79,80]. This contraceptive must be delivered daily for at least 50 days in order to inhibit production of litters for around three successive breeding rounds, as shown in captive studies with Norway rats [81,82]. In this species ContraPest decreased the weight of reproductive organs but had no effect on adrenal, kidney, spleen and liver weights compared to control animals [82]. The efficacy and potential side effects of ContraPest on free-living rats has not been reported, as only information on the efficacy of ContraPest combined with a rodenticide is available [78].

EP-1, based on synthetic steroids, has been proved to inhibit the fertility of males and females of many rodent species in China, Tanzania, Zambia, and Indonesia [83,84,85,86,87] in captivity and field trials. In several species, a treatment period of about 7 days in laboratory studies or a single baiting with EP-1 in field conditions, are sufficient to induce infertility [87]. In females the most common response to EP-1 is an enlargement of the uterus which result in reduced conceptions and/or litter sizes. In males EP-1 inhibits the function of the testis, epididymis and seminal vesicles for different periods of time depending on the dose [75,87] and in both sexes the effects are temporary and fully reversible. Side effects of these synthetic hormones in rodents include production of smaller pups in striped field mice (*Apodemus agrarius*) dosed with EP-1 [88] and in Brandt’s vole (*Lasiopodomys brandtii*) treated with quinestrol [74]. EP-1, widely tested on many rodent species, has not been used in field trials in Europe and its use might not be acceptable until the potential effects of the hormones on the food chain and on the environment have been assessed.

Nicarbazin (NCZ) is the active ingredient of oral contraceptives commercially available for pigeons and registered as a veterinary medicine in Italy (Ovistop) and in Belgium (R-12) and as a biocide in the USA (Ovocontrol), where it is also registered for use with Canada geese (*Branta canadensis*) [26,29,77]. NCZ, traditionally used to manage coccidiosis in chickens, disrupts the membrane between the egg albumen and the yolk, thus preventing the development of an embryo [89]. NCZ must be fed continuously before and during egg-laying to be effective [26]. As once ingested nicarbazin dissociates into two inactive compounds, it is unlikely to affect birds of prey feeding on treated pigeons [77]. Because NCZ is rapidly cleared from the body, once consumption of NCZ-treated bait ceases, its contraceptive effect is reversible. 

## 3. Delivery Methods

Contraceptives are delivered to wildlife in two ways: by intramuscular injection, administered manually or remotely, and by bait. The cost of using injectable contraceptives, such as GnRH- or ZP-based vaccines is relatively high as this method requires capture, injection and release for first vaccination and in some instances re-capture for administration of booster doses to increase the duration of the contraceptive effect. Booster doses of some contraceptives can be delivered remotely by dart gun [45,90]. Remote dart delivery of immunocontraceptive vaccines may prove the most efficacious method for relatively small, discrete populations with no or minimal immigration, such as on golf courses, urban parks and reserves [45,91,92,93]. Distance-adjustable CO_2_-powered dart rifles have been employed to fire 2–3-mL syringe-darts at ranges of 40 m into the hindquarter of large mammals. 

The advantages of these delivery systems include that they target individual animals, they can administer individually tailored doses based on a body weight, and they can be used to avoid both the welfare and economic costs of trapping. Potential disadvantages include identification of previously vaccinated individuals, dose regulation and incomplete intra-muscular injection [94,95]. Evans et al. [96] suggested that this methodology needs to be refined before remote delivery of a contraceptive vaccine becomes a usable technique. When contraceptives are administered remotely it is important that animals can be individually identified, for instance by natural marks such as coat colour or horn shape, or by equipping them with ear tags at first vaccination [92,97]. Some studies based on remotely delivered contraceptives have used darts that leave a paint mark for several weeks on treated animals so that operators can avoid re-treating these individuals [90].

In Europe injectable contraceptives have been used for small, isolated populations, particularly in areas where culling is publicly opposed or illegal. This is the case for a population of feral goats in Wales, successfully treated with GonaCon [65] and for feral horses in the Danube Delta in Romania, treated with a PZP-based immunocontraceptive vaccine [46]. The horses in the Danube Delta study showed a behavioural aversion to darting, with first darts being delivered from a distance of about 25–30 m but the following darts from 40–60 m. 

All the oral contraceptives currently available for wildlife have the potential to affect reproduction of non-target species. Hence, they must be delivered through methods that minimize consumption by non-target species. Some specificity can be achieved by placing the bait in active burrows [98], but in many instances consumption by the target species is achieved by using custom-designed bait delivery devices. Examples include bait boxes for rats which limit access to contraceptives by non-target species [72], bait distributors of nicarbazin, designed for urban pigeons and used in several European cities [99], systems conceived for delivering baits to wild boar and feral pigs such as the BOS (Boar Operated System), tested in the UK, in the US and in Italy [100,101,102] and hoppers used in the UK to deliver baits to Eastern grey squirrels [103].

## 4. Feasibility, Costs and Public Attitudes

Many laboratory or captive studies have tested the effect of contraceptives on individual animals, fewer studies have assessed these effects on free-living wildlife and even fewer have investigated the impact of this method at population level and in particular whether and for how long fertility control may reduce population size or impact. Understanding how fertility control affects population dynamics and animal behaviour is crucial for evaluating the effectiveness of this method at reducing population size or growth or at decreasing the environmental and economic impacts associated with wildlife [29]. This knowledge is also essential to plan the effort and time required to achieve the desired outcome. In an ideal world, fertility control would affect only reproduction; in reality, it may also affect survival, social and spatial behaviour, and lead to changes in immigration and emigration. 

When the local number of animals is reduced, compensatory density-dependent processes may cause the population to return to its previous level. For instance, in populations of mice and rabbits, a compensatory response in female productivity was recorded, although this did not offset the effects of sterilisation when 60–80% of the females were made infertile [104,105]. Long-lived species are likely to be easier to manage using fertility control than smaller, shorter-lived ones because a lower proportion of the population must be rendered infertile at any time [106].

In terms of time taken to achieve a reduction in population size, culling is always more efficient than fertility control as infertile animals remain in the population for their natural lifespan. However, fertility control might be more efficient than culling if infertile individuals maintain sufficient density-dependent feedback on recruitment [107]. In addition, in instances where diseases are transmitted from mother to offspring, such as brucellosis in bison, fertility control could be employed to reduce such transmission [57].

In Europe the use of fertility control via injectable immunocontraceptives has been proposed to complement vaccination against bovine tuberculosis in badgers and thus to reduce recruitment of new susceptible offspring into these populations [64].

Several models focused, for instance, on badgers, wild boar, deer and grey squirrels, have suggested that fertility control can be used to complement other methods to reduce populations, once a target density has been achieved [108,109,110,111]. For wild pigs adding fertility control to culling, was predicted to reduce abundance substantially more than culling alone, particularly in areas open to immigration, because these populations could not be reduced substantially by culling alone [112]. In some instances, fertility control has been employed to prevent population growth rather than to decrease population size. Druce et al. [113] introduced the concept of ‘rotational immunocontraception’, showing that rotating elephants treated with a reversible immunocontraceptive increased the time between births for a single female and simultaneously gave all animals the possibility to participate in reproduction. 

The cost of fertility control to manage free living wildlife populations depends on a number of factors including time and manpower estimated to achieve target population reduction, cost of materials such as contraceptives themselves, anaesthetics, traps or bait dispensers to deliver contraceptives. These costs should be weighed against other methods of population control as well as against the economic and environmental impact of a particular species. For instance, a model comparing the effectiveness of fertility control and culling to reduce grey squirrels in the UK suggested that, when applied to the low-density populations following short-term culling, population reduction using contraceptives could be achieved within the same timescales as continuous culling alone but with substantially lower costs [111]. 

In a European context, manpower, which often represents the main cost for delivering contraceptives to wildlife, could be offset by involving volunteers and community groups in implementing fertility control programs. Citizen science and volunteer engagement with wildlife management have increased dramatically in the recent decades [114] and volunteers play a significant role in implementing methods to manage wildlife. For instance, in the USA a partnership between the Bureau of Land Management (BLM), responsible for the protection and management of wild horses and donkeys on public lands, and non-government organizations has been very successful in delivering contraceptives to these animals. In populations of 50–150 free-roaming equids, immunocontraceptives are delivered remotely to individually identified animals. The work is conducted with BLM approval, but staffed extensively by non-government local residents committed to a specific herd [90]. This model has been used in two successful projects on fertility control for deer. A key benefit of this model is that the public becomes partners with the governmental agency and this commitment minimizes conflicts among local stakeholders [90]. 

Similarly, volunteer groups in Europe have been key to monitor reinvasion by grey squirrels on Anglesey in Wales [115]. It is important to note that, both in Anglesey and in other parts of the UK, community groups involved in grey squirrel control mentioned they would prefer alternatives to lethal population control [115,116]. This attitude reflects the results of a recent survey on public acceptability of population control methods for grey squirrels that found that traditional lethal methods were regarded as least acceptable, whilst contraception was the preferred method, supported by 63% of the 3758 respondents [117]. 

In Europe fertility control has also been suggested for managing wildlife disease, as an alternative to culling, where lethal methods are uneconomic, ineffective or are not publicly supported. For instance, in the UK where badgers are a wildlife reservoir of bovine tuberculosis (bTB), the effectiveness of an injectable vaccine against bTB, administered to free-living badgers to reduce transmission of bTB, could be substantially increased by adding fertility control to reduce the recruitment of susceptible young animals [108,118]. In this scenario, the economic cost of using injectable immunocontraceptives would be minimal, as the animals will be trapped and vaccinated anyway, thus increasing the overall efficacy of disease control.

## 5. When to Use Wildlife Fertility Control? A Framework for Decision Making

Fertility control should be considered in all instances when lethal control is regarded as ineffective, inefficient, unacceptable, environmentally harmful, unfeasible, illegal or any combination of these elements. Examples of contexts for applications of wildlife fertility control include charismatic species, such as many feral livestock, urban or peri-urban environments, national parks and areas where culling is illegal [26,119]. 

Massei and Cowan [29] provided a framework to guide decisions on the suitability of fertility control to manage wildlife impacts and recommended a staged approach formulated as a decision tree. The approach involves identifying and testing a contraceptive for the species to be managed, initially in captivity and subsequently in field studies, addressing whether the effects of the contraceptive on behaviour and welfare are acceptable, assessing whether the contraceptive could be delivered in field conditions, evaluating whether the desired goal could be achieved in a set time frame, assessing costs and sustainability of implementing this method and taking into account public attitudes.

When considering fertility control for a new species, assuming a contraceptive has been shown to be effective and safe in this species, many questions should be answered before implementing this method (Figure 1). Key points to address concern public support, legal issues (as many contraceptives could be imported and used only under experimental research permits), the availability of experienced staff and resources to deliver contraceptives, often covering multiple years. Ideally, staff should be experienced in capture, immobilization and release techniques (if needed to administer contraceptives), in methods to assess animal welfare, in using devices for bait-delivered contraceptives and in monitoring bait uptake. Other points to consider when discussing fertility control for a specific context include the costs, so that an adequate budget can sustain the work, and the practical aspects of delivering contraceptives to a pre-determined proportion of the population without affecting non-target species. In addition, robust methods to assess population size and/or impact before, during and after the application of fertility control must be in place, with staff trained to measure the impact of this intervention.

Modelling can be used a priori to determine the proportion of the population that must be rendered infertile to achieve the target reduction in population size or a reduction in impact. Research questions to be considered to explain the effect of fertility control on population size or impact concern the potential effect of fertility control on survival, emigration or immigration, and on the reproductive output of non-treated animals. For instance, will a larger proportion of non-treated animals start to reproduce, or will they have larger or more litters per year? Most projects generally answer one or a few of these questions, although all these aspects are crucial for making decisions about the effective use of wildlife fertility control in a particular context.

Defining the objectives of the program and the timeline is also essential as fertility control could be considered for instance for decreasing population size or growth, reducing the impact of wildlife or for decreasing the prevalence of a disease. This is complicated by the fact that population size and environmental or economic impacts are not always linearly related, i.e., a reduction in population size is not necessarily followed by a proportional decrease in damage [120,121]. Additional research on the relationship between local wildlife numbers and associated economic and environmental impacts is required to establish density thresholds, which will vary for different species, below which wildlife and people can coexist. It is also important to stress that, particularly for long lived species, infertility will not lead to a sudden decrease in impact as all the animals rendered infertile may still live for several years, until their natural death. 

Field trials are likely to be expensive as establishing the impact of an injectable contraceptive at population level often requires estimating population size before and after the treatment with the contraceptive, treating tens of free-living animals and monitoring their reproductive output in field conditions or at least the recruitment of juveniles in the population. However, field trials are crucial to establish effectiveness, cost and the feasibility of fertility control to manage specific populations

## 6. The Future: Why Should Europe Invest in Wildlife Fertility Control?

There are many reasons for European Member States to invest in developing contraceptives for wildlife (Figure 2). The first is that, as wildlife impacts increase, culling is consistently proving ineffective in reducing the number of overabundant native and non-native wildlife species and their impacts. An example is provided by wild boar: the number of hunters, that for decades have been the main cause of mortality for this species, is stable or declining in most European countries, whilst the number of wild boar culled is increasing dramatically [7]. For instance, the number of wild boar culled by hunters in France and in Germany rose from circa 100,000 per year in the early 1980s to more than 500,000 animals three decades later [7]. Whilst hunters play an important role in reducing the number of ungulates across Europe, the fact that deer and wild boar are increasing means that the impact of these species can only escalate [4]. For non-game, non-native invasive wildlife, such as raccoons, coypu, parakeets and squirrels, the interest or willingness to cull are even lower than those for game species and thus the number and range of these animals will continue to grow. Fertility control offers an alternative approach and would add another method to the toolbox of options to resolve wildlife impacts.

The second reason for investing in developing wildlife contraceptives is that public attitudes on human–wildlife interactions are shifting from conflict management to coexistence, with stakeholders demanding alternative solutions to culling [31,122]. In some circumstances, fertility control may have inherent advantages over culling [118]. These advantages include: 1. infertile individuals that remain in the population may contribute to density-dependent processes that limit the number of animals reproducing [123]; 2. fertility control could reduce problems associated with breeding activity such as burrow and nest construction or expansion and relative nuisance behaviour [124]; 3. fertility control might encourage breakdown of breeding pair bonds, arising from reproductive failure, in species exhibiting site and mate breeding fidelity, which would reduce the local breeding population; 4. compared to fertility control, culling is more likely to result in increased movement outside the area where culling occurs [125], potentially increasing the likelihood of vehicle collisions and disease transmission.

The third reason for Europe to invest in developing contraceptives is linked to wildlife diseases. Contraceptives could work synergistically with disease vaccines to manage wildlife disease reservoirs such as bovine tuberculosis in the European badger, rabies in foxes (*Vulpes vulpes*) or classical swine fever in wild boar. For instance, a model suggested that when adding fertility control to rabies vaccination for free-roaming dogs [126], the control rate and duration required for rabies eradication were reduced by about half, supporting similar predictions from fox rabies models [127]. Fertility control could assist disease eradication by lowering birth rates so that fewer susceptible offspring are added to the population. This provides two benefits: the proportion of vaccinated animals remains high, and the population is maintained at a lower density, thus reducing the risk of reinfection. In the context of wildlife diseases, culling may result in increased movement and contact between individuals, thus increasing the risk of disease transmission, whilst fertility control would be expected to cause less social perturbation than culling. Fertility control could reduce the transmission of disease from mother to offspring which is important for disease maintenance in a population [57]. In addition, by reducing the physiological burden of gestation, lactation and the energetic costs of reproduction (for instance due to males fighting other males for access to females), fertility control may enhance the physical condition of animals, thus potentially reducing their susceptibility to diseases. 

The fourth reason for Europe to invest in fertility control is the increased public awareness of the environmental impact of some toxicants, coupled with rodenticides resistance across several European countries that is already limiting the impact of these toxicants on rodent populations [128]. Despite strategies developed to reduce the effects of rodenticides on non-target species [129], many problems remain. For instance, in Spain, rodenticide use during vole outbreaks caused the death of non-target wildlife (game species and endangered raptors) as well as heated conflicts between stakeholders. As rodents feed migratory raptors worldwide, the impact of rodent management on biodiversity extends beyond the regions that apply anticoagulants [130]. In a review on the status of fertility control for rodents, Jacoblinnert et al. [22] concluded that, compared to anticoagulant rodenticides, contraceptives have the potential to deliver a higher degree of humaneness and therefore, gain more public acceptance. 

The fifth reason for Europe to promote investment in wildlife fertility control is that this method could be used for small, isolated populations of wildlife and livestock that cannot be managed through lethal control. An example of these contexts is given by the rewilding movement which is growing across rural landscapes in many Member States [131]. Rewilding focuses on the restoration of self-sustaining ecosystems in which human intervention is minimized. In these ecosystems, free roaming horses and cattle are used to replace the now extinct native large herbivores such as mammoths and aurochs. However, when these populations grow and risk starvation, some human interventions are required. The risk of non-intervention was exemplified by one of the first rewilding projects in the Netherlands where a scheme to rewild the marshland in the Oostvaardersplassen ignited public protest after deer, horses and cattle reached such local densities that animals were starving and had to be culled [132,133]. Another example of potential use of fertility control is the rewilding project in the Faia Brava Reserve in Portugal, where horses and cattle receive supplementary feeding, particularly in harsh winters, but also where cattle are sold for meat and horses are adopted out to decrease local numbers [134]. As with several other free living livestock populations, there is only a limited proportion of the population that can be adopted out [62,65]. Once this limit has been reached fertility control in these areas might play an important role in reducing population growth. In rewilding areas is it likely that some human intervention will be necessary to reduce population size of large herbivores and to prevent these animals from starving. As many of these areas are under public scrutiny, fertility control for these contexts is likely to be preferred to culling.

The sixth reason for Europe to invest in fertility control is that volunteers and community groups have multiplied in recent decades and have expressed their willingness to collaborate with conservation and with wildlife management projects. This enormous reserve of manpower should be used to implement otherwise costly fertility control programs based on injectable contraceptives, particularly as several studies found that volunteer and community groups prefer fertility control to culling [115,116,117]. Using fertility control to reduce wildlife impacts also would offer the means of engaging local communities and promoting wildlife stewardship.

The seventh reason for investing in developing wildlife contraceptives in Europe is that in several contexts culling is illegal or too dangerous to be carried out due to the risks of harming people and companion animals. This is certainly the case for using firearms in urban, highly populated areas but it can also apply to situations where removal is carried out by trapping and dispatching or by trapping and translocations. In these instances, people opposed to culling may interfere with traps; translocations may lead to translocated animals travelling far to return to their natal areas, whilst suffering from malnutrition and increased risk of vehicle collisions, “re-offending” in the new area and spreading pathogens that do not occur in the new area [135,136,137].

The eighth reason for Europe to promote investment in wildlife fertility control is based on the increasing number of mathematical models (quoted above) suggesting that this method could be used, alone or to complement other wildlife management options, to reduce local wildlife abundance and/or impacts.

## 7. Conclusions

This review highlighted that some wildlife contraceptives are already available for field applications and for specific contexts, although none of these products is currently available in European countries, with the exception of oral contraceptives for pigeons. In Europe the use of injectable contraceptives for wildlife is currently limited to relatively small, isolated populations, such as feral livestock, charismatic species, but also situations in which culling is not legal nor safe, such as urban and peri-urban areas or regional and national parks where culling might not be acceptable. 

A key area of investment for Member States interested in meeting public demands for alternative methods to culling is the development of oral contraceptives which would widen the spectrum of practical applications of wildlife fertility control. Developing oral contraceptives is challenging and expensive as the registration of these drugs in Europe may cost several millions of euros and take a few years. Although these costs appear substantial, the economic, social and environmental benefits of bringing to the market oral contraceptives vastly exceed the actual cost of the impact of some wildlife species. More research is also needed on cost-effective systems to deliver oral contraceptives to target species, on assessing the effects of oral contraceptives on the food chain, and on field testing of the impact of fertility control at population level. We do not know yet to what extent fertility control could complement or substitute lethal control but the scale, variety and growth of wildlife impacts throughout Europe suggest the time to find new tools to mitigate these impacts is now. 

## Figures and Tables

**Figure 1 animals-13-00428-f001:**
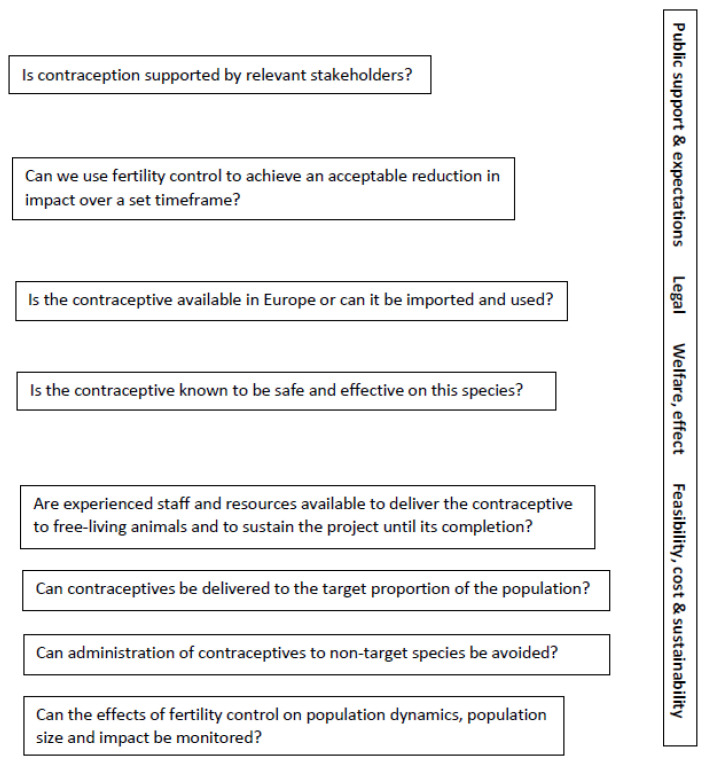
A framework to guide decisions about the use of fertility control to mitigate wildlife impacts. Illustrated are questions that should be considered before implementing a fertility control plan for free-living wildlife. If the answer to all the questions is “yes”, the plan provides a framework for assessing whether fertility control will be likely to meets the stated goals. If the answer to any question is “no”, fertility control should not be implemented until the issue has been resolved.

**Figure 2 animals-13-00428-f002:**
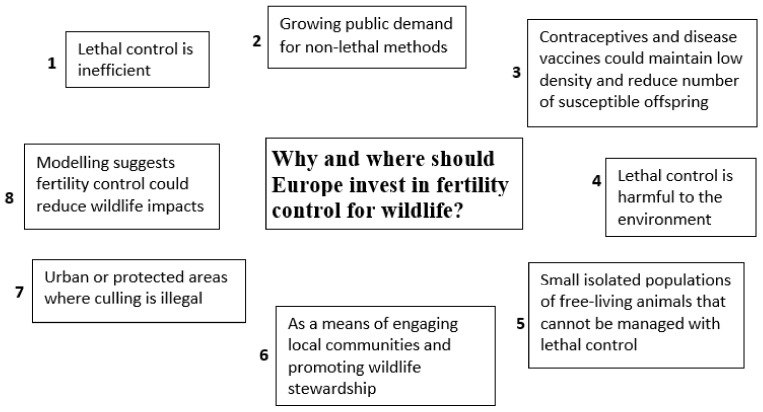
Reasons and contexts for Europe to invest in research on wildlife fertility control to mitigate wildlife impacts. ”Wildlife” includes wild mammals and bird species and feral livestock.

## Data Availability

Not applicable.

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
