# Peer review of "Fertility Control for Wildlife: A European Perspective"

_animals, 2023, doi:10.3390/ani13030428_

Round 1

Reviewer 1 Report

This is a very good, and in my opinion necessary, review of the available methods for fertility control in wildlife. Conflicts caused by the overgrowth of native and/or alien wild populations are increasing. However, culling has not been proven to be an effective method in the long term. For this reason, it is necessary to propose new methods of population control, such as the use of contraceptives.

This review begins with an introduction explaining the need for these methods, followed by a description of the different methods currently available, their administration, and their costs. The manuscript is written in a very complete and easy-to-read way, it shows that it has been worked on, and a high percentage of the references used are recent (<10 years). Despite being a "simple" review, it raises serious questions that in certain circumstances could even be controversial. But that's what science is for, to answer questions with new questions.

As the author indicates in the text, most of these methods have not been well studied in wild species, and require repeated administration every X months or even years. The first issue that must be taken into account is the identification of the treated animals in order to recapture them in the future and administer the next dose. This identification greatly complicates the correct treatment of wild animals and would make fertility control programs more expensive. In addition, another question that arises is the possible repercussions of contraception within populations (behavior of individuals during the mating season, courtship, etc.) and between populations of different species. Could there be repercussions at the ecosystem level?

At present, the problems due to the overpopulation of wildlife are treated by culling, a very effective technique to reduce the population promptly, as the author indicates. However, by sacrificing all these animals, the ecological niche they occupied will be available for other individuals (generally of the same species) to occupy it. Therefore, in the long term, the problem is not solved. With the administration of contraceptives, these ecological niches could be kept occupied by infertile individuals.

In summary, this review, although it may be controversial, is currently very useful and much needed.

I only have some small corrections and suggestions for the author:

- L42, 80, 285, 294, 356 (x2), 491, 513, 561: please remove double spaces.

- L89: a period is missing at the end of the sentence.

- L99: please add a colon before listing the targets. Do the same on line 459.

- L102: in the previous objectives you have not started the sentences with a capital letter, please, homogenize. The same thing happens on line 465.

- 2.Wildlife contraceptives: this is the longest section, and the most important, of the review. To make it easier to read, I recommend dividing the text into two subsections: 2.1. Immunocontraceptive vaccines (L127-218) and 2.2. Oral contraceptives (rest).

- L126: please, add a reference on efficacy factors of immunocontraceptives.

- L131-132: it seems that there was a paragraph break that shouldn't... The same thing happens in lines 488-489

- L134-141: this information could be included in a table to facilitate comparison between them.

- L142-154: Apart from the white-tailed deer, are there studies carried out specifically on wildlife? I suggest continuing this paragraph with the paragraph on lines 164-172 and then describing the negative effects.

- L160-163: in which species have these negative effects been seen? Move this sentence to line 157.

- L190: I suggest moving the information from lines 203 to 212 here and continuing with the negative effects, the same as with the PZP-based immunocontraceptives.

- L219-257: except for EP-1, no side effects or duration of infertility are described for oral contraceptives. Are there studies about it? I think this is important information that should be reflected in the review. If there are no studies on this, please indicate so.

- L369: please write well the number of respondents (3,758).

- Figure 1: I suggest adding one more question: "Can it affect other collateral wild species?"

- The conclusions are too long. I suggest shortening them as much as possible. A conclusion should be clear and concise. 

Author Response

Please  find my replies in Italics

In summary, this review, although it may be controversial, is currently very useful and much needed.

I only have some small corrections and suggestions for the author:

  1. L42, 80, 285, 294, 356 (x2), 491, 513, 561: please remove double spaces. Done.
  2. L89: a period is missing at the end of the sentence. Done.
  3. L99: please add a colon before listing the targets. Do the same on line 459. Done.
  4. L102: in the previous objectives you have not started the sentences with a capital letter, please, homogenize. The same thing happens on line 465. Done.
  5. Wildlife contraceptives: this is the longest section, and the most important, of the review. To make it easier to read, I recommend dividing the text into two subsections: 2.1. Immunocontraceptive vaccines (L127-218) and 2.2. Oral contraceptives (rest). Done
  6. L126: please, add a reference on efficacy factors of immunocontraceptives. Done
  7. L131-132: it seems that there was a paragraph break that shouldn't... The same thing happens in lines 488-489 Corrected.
  8. L134-141: this information could be included in a table to facilitate comparison between them. This information is only a few lines about three formulations, which will make a table very small. For this reason, I left it as text.
  9. L142-154: Apart from the white-tailed deer, are there studies carried out specifically on wildlife? The papers quoted refer to several species, i.e. white-tailed deer, horses and feral donkeys I suggest continuing this paragraph with the paragraph on lines 164-172 and then describing the negative effects. Agreed and moved the paragraph as requested.
  10. L160-163: in which species have these negative effects been seen? Move this sentence to line 157. Done and added examples of species.
  11. L190: I suggest moving the information from lines 203 to 212 here and continuing with the negative effects, the same as with the PZP-based immunocontraceptives. Done
  12. L219-257: except for EP-1, no side effects or duration of infertility are described for oral contraceptives. Are there studies about it? I think this is important information that should be reflected in the review. If there are no studies on this, please indicate so. Done, I have added this information.
  13. L369: please write well the number of respondents (3,758). Done
  14. Figure 1: I suggest adding one more question: "Can it affect other collateral wild species?" Done. I have added to Fig. 1 the question “Can administration of contraceptives to non-target species be avoided?”
  15. The conclusions are too long. I suggest shortening them as much as possible. A conclusion should be clear and concise. Done, I have shortened the conclusions to ~ 50% of the original text.

Reviewer 2 Report

This is an interesting and important review.  It is generally well written though I have suggested the removal of numerous superfluous commas which would improve the flow of the expression.  There are two sections where I am suggesting that improvements could be made.  They are where the author mentions the relationship between animal density and damage.  This aspect of the review is fundamental and it has been researched elsewhere.  I have included a reference which I hope will be helpful.  Secondly I think there should be some acknowledgement of the current importance of lethal methods such as anticoagulant toxicants and the methods that can be used to reduce their undesirable side effects.  Again I have included a reference.  All other comments are in the attachment.

Author Response

Please see below my replies in blue Italic

This is an interesting and important review.  It is generally well written though I have suggested the removal of numerous superfluous commas which would improve the flow of the expression.  There are two sections where I am suggesting that improvements could be made.  They are where the author mentions the relationship between animal density and damage.  This aspect of the review is fundamental and it has been researched elsewhere.  I have included a reference which I hope will be helpful.  Secondly I think there should be some acknowledgement of the current importance of lethal methods such as anticoagulant toxicants and the methods that can be used to reduce their undesirable side effects.  Again I have included a reference.  All other comments are in the attachment.

Abstract

Generally informative but I would recommend giving more of the conclusions of the review. For example from line 33 where the author says she will give the reasons why Europe should invest in these methods, she couldtell the reader that she has found seven reasons she found and even make a broad statement about what she has concluded about the efficacy of fertility control. It would be helpful if the abstract also indicated that thisreview contained a decision making framework to help practitioners decide where and when to consider fertility control as a management tool.

Introduction

  1. Line 63. The sentence containing ‘…in Germany financial losses due to crop damage resulted in hundreds of 63 millions of euros’ would be more clearly worded something like ‘…in Germany financial losses due to crop damage amounted to 63 millions of euros…’ Amended.

2.There is also an inconsistency in this paragraph between the way the amount of currency is reported.It is firstly written as 63 millions of euros and subsequent mentions of a monetary amount are written with the symbol for that currency. Done.

3.Line 90. Remove comma after (EU). Done.

Wildlife Contraceptive

  1. Line 111. I suggest that ‘Zoo’ should be plural ‘Zoos’. Amended.
  2. Line 208. The systematic name for badgers has already been presented and does not need to be repeated here. Amended.
  3. Line 219. Remove comma after ‘At present…’ Done.
  4. Line 223. The systematic name for Norway rats has already been presented and does not need to be repeated here. Also ‘…which is a combinations of two…’ should be singular ‘combination’. Amended.
  5. Line 233. Remove comma after daily. Done.
  6. Line 241. In the sentence beginning ‘In females,…’ remove both commas. Done.
  7. Line 242. Remove comma after males. Done.

11, Line 243. The sentence containing ‘…for different periods of time in relation to the dose…’ Would be better worded ‘…for different periods of time depending on the dose...’ Amended.

  1. Line 248. Sentence beginning ‘Nicarbazin (NCZ) is the active ingredients of oral contraceptives…’The word ‘ingredients’ should be singular, ‘ingredient’. Amended.
  2. Line 249. Remove comma after pigeons. Done.

Delivery Methods

14, Line 267. CO2 should be written CO2. Amended.

  1. Line 277. Remove comma after remotely. Done.
  2. Line 281. Remove comma after animals. Done.
  3. Line 282. Remove coma after Europe. Done.

Feasibility, costs, public attitudes

  1. Line 317. There is a problem with the citation at the end of the sentence. Amended.
  2. Line 318. In the sentence beginning ‘In terms of…’ remove both commas. Done.
  3. Line 324. Remove comma after Europe. Done.
  4. Line 329. Remove comma after populations. In the next sentence beginning ‘For wild pigs…’ remove the first two commas. Done.
  5. Line 332. Remove comma after instances. Done.
  6. Line 361. Remove comma after agency. Done.
  7. Line 368. Remove comma after squirrels. Done.
  8. Line 371. Remove comma after Europe. Done.

When to use wildlife fertility control? A framework for decision making

  1. Figure 1. I would suggest that another question needs to be added. Is contraception acceptable to relevant stakeholders? Done, I have added this question as the first one and I also added a “side bar” to Figure 1, to link the questions to various issues such as public support, welfare, feasibility etc. Further, I would also suggest that the first question be modified. Wildlife managers will not be interested in simply reducing the population size. They will want to reduce the impacts that the particular population is causing. So I believe the first question should be modified to ‘Expectations and goals: based on the known impact can we use fertility control to achieve an acceptable reduction in impact over a set timeframe?’ It may well be possible to reduce the population size without reducing the unwanted impact, and that has been canvassed in this manuscript. Done, I have now changed this question to “Can we use fertility control to achieve an acceptable reduction in impact over a set timeframe?” and I am grateful for this suggestion.
  2. Line 421. Remove comma after essential. Done.
  3. Line 423. Sentence beginning ‘This is complicated by…’ This is a very important point which needs to be emphasized. One way to do that would be to provide one or more authoritative references. A good place to start would be Hone, J. 2007. Wildlife Damage Control. CSIRO Publishing, Collingwood Victoria, Australia. I point out his table 2.2 beginning on page 18 which cites numerous examples of this point. Done, I have added this reference and added a second one (Krull et al. 2016).
  4. Line 426. The sentence beginning ‘Very little research has been carried out…’ The above reference demonstrates this statement is not true. This section should be modified to reflect the research that does exist. Please see point 28. I have also modified this sentence that now reads “Additional research on the relationship between local wildlife numbers and associated economic and environmental impacts is required to establish density thresholds, which will vary for different species,..”
  5. Line 432. Remove comma after present. Done.
  6. Line 475. Remove comma after rates. Done.
  7. Line 481. Remove comma after offspring. Done.
  8. Line 487. The paragraph beginning on this line is very important. The author should note that currently for rodents control over an area larger than about a football field is now only possible using anticoagulants. Hence numerous strategies have been developed to reduce the unwanted side effects of their use. See for example P A. Castaño, K J. Campbell, G S. Baxter, V Carrion, F Cunninghame, P Fisher, R Griffiths, C C. Hanson, G R. Howald, W J. Jolley,B S. Keitt, P J. McClelland, J B. Ponder, D Rueda, G Young, C Sevilla, N D. Holmes. 2022. Managing non-target wildlife mortality whilst using rodenticides to eradicate invasive rodents on islands. Biological Invasions DOI: 10.1007/s10530-022-02860-0. These mitigation strategies should at least be acknowledged here. Done.
  9. Line 504. Remove comma after starvation. Done.
  10. Line 506. Remove comma after Netherlands. Done.
  11. Line 513. Remove comma after reached. Done.
  12. Line 515. Remove comma after areas. Done.

Conclusions

  1. Line 541. Remove comma after Europe. Done.
  2. Line 545. Remove comma after countries. Done.
  3. Line 547. Remove comma after expensive. Done.
  4. Line 556. Remove comma after contraceptives Done.

Reviewer 3 Report

The background and review of what methods exist for contraceptives for wild animals are good. But the model with questions presented are not structured or logical and needs to be improved before publication. Also the last section with a figure with 8 different reasons for using this in Europe and the text describing 7 different reasons. This needs to be improved and synchornised.

Author Response

Please find below my replies in blue Italic.

1. The background and review of what methods exist for contraceptives for wild animals are good. But the model with questions presented are not structured or logical and needs to be improved before publication. Also the last section with a figure with 8 different reasons for using this in Europe and the text describing 7 different reasons. This needs to be improved and synchornised. I have re-written parts of this session and matched the order of the questions in the text with that presented in Figure 2.

2. Li 45. In what direction do you see this impact on biodiversity? The other list in effect are directed from the wildlife towards human interests. Good point: to answer this, I have substituted “human-wildlife conflicts” in the original manuscript with “wildlife impacts” throughout the text.

3. Line 56 are the native or non-native species a threat to biodiversity?Yes, some wildlife species that become “overabundant”, like the wild boar and deer, that are not regulated by natural predators, may affect biodiversity and species composition. I did not give examples as I felt these would make the Introduction too long.

4. Line 101. As the model is presented now I cannot find a logical structure. You have listed a number of questions both in figure 1 and in the text but no relationship or order between these questions. You need to change that in order to fulfil this part of your aim. Done,  I have changed this figure and part of the text to address this point.

5. Line 139. This looks strange. Shouldn’t be “three “the “iii “? Amended.

6. Line 358. How is the people who delivered the contraceptives trained and how is the welfare of the animals checked? I have added a sentence to explain how people who deliver the contraceptive should be trained. For animal welfare, please see next point.

7. Line 394. I miss a discussion about how the welfare of the wild animals that we are injecting vaccines in could be affected I don’t see that in the questions you ask here. I have now mentioned animal welfare both in the text (…assuming a contraceptive has been shown to be effective and safe in this species…) and in Figure 1. Adding a discussion on methods to assess the impact of fertility control on animal welfare, or indeed methods to estimate local densities or impacts was beyond the scope of this paper.

8. Line 406. Are the questions in the model in a certain order or equally important? If the answer to one of these questions is no what happens then? Could they somehow be arranged in a way that you answer them in a specific order in you give different options depending on the answer? Like a taxonomic key for species. Please see my answer to point #3. The legend of Figure 1 now mentions that if the answer to all the questions is “yes”, the plan provides a framework for assessing whether fertility control will be likely to meets the stated goals. If the answer to any question is “no”, fertility control should not be implemented until the issue has been resolved.”

9. Line 412-416. Why are not these questions in your framework in figure one? This part needs structure to be able to work as a model for decision-making. One of these questions, related to stakeholders, is now part of Figure 1. Questions related to survival, emigration and immigration are part of the last question on effects of fertility control on population dynamics and on impact. In the text I have listed research questions that could be considered to explain the effect of fertility control on population size or impact.

10. Line 438. In the figure you have eight boxes with reasons and in the text you list seven reasons. How are they related? and can you number the boxes in the same way to make it easier for the reader to see the connection between figure and text please. Amended, with thanks to the reviewer for noting the discrepancy. The (now) 8 reasons for Europe to invest in wildlife fertility control reported in Figure 2 are discussed in the text in the same clockwise order.

11. Line 536. In your conclusion I would like to see conclusions from what you have already reported above. In this paragraph you continue the discussion about what Europe needs to invest their research in. You take up new areas here and develop them instead of concluding from the suggestions and model that you have outlined about. Please rewrite the conclusion. I have shortened considerably  and partly re-written the conclusions.

Round 2

Reviewer 2 Report

The author has taken on board my concerns and recommendations from the first review.  I judge the second submission to be an improvement over the first.  I believe this will be a valuable contribution for practitioners and decision makers, and discussion starter for many interested parties.

Author Response

I have checked the English language and style and corrected fine/minor spelling mistakes

Reviewer 3 Report

I would like you to add numbers to the different squares in Figure 2 that correpsonds to the 8 reasons listen in the text.

Author Response

I have checked the manuscript for English language and style and for fine/minor spelling mistakes